# Prevalence of polypharmacy and drug interactions in geriatric patients: A cross-sectional study from India

Umaima Farheen Khaiser[1], Rokeya Sultana[2], Ranajit Das[3], Saeed G. Alzahrani[4], Madhumitha Kannan[5], Mohammed Khairt Newigy[6], Mohammed Sharique Ahmed Quadri[7], Mujeeb Ahmed Shaikh[7], Farha M. Shaikh[7], Mohammad Fareed[7,8*]

1 Department of Pharmacognosy, Department of Pharmacy Practice, Yenepoya Pharmacy College and Research Centre, Yenepoya (Deemed to be University), Mangalore, Karnataka, India, 2 Department of Pharmacognosy, Yenepoya Pharmacy College and Research Centre, Mangalore, India, 3 Division of Data Analytics Bioinformatics and Structural Biology, Yenepoya Research Centre, Mangalore, India, 4 Department of Family and Community Medicine, College of Medicine, Imam Mohammad Ibn Saud Islamic University (IMSIU), Riyadh, Kingdom of Saudi Arabia, 5 Saveetha Medical College and Hospital, Saveetha Institute of Medical and Technical Sciences (SIMATS), Chennai, Tamil Nadu, India, 6 Department of Anesthesia Technology, College of Applied Sciences, AlMaarefa University, Diriyah, Saudi Arabia, 7 Department of Basic Medical Sciences, College of Medicine, AlMaarefa University, Diriyah, Riyadh, Saudi Arabia, 8 Research Center, Deanship of Scientific Research and Post-Graduate Studies, AlMaarefa University, Diriyah, Riyadh, Saudi Arabia

* fareed.research@gmail.com

## Abstract

### Background

Polypharmacy and potential drug-drug interactions (pDDIs) are major challenges in the management of elderly patients with multiple comorbidities, often leading to adverse drug events and increased healthcare burden.

### Aim

This study aimed to determine the prevalence and patterns of polypharmacy and pDDIs among elderly patients attending Yenepoya Medical College and Hospital, Mangalore, India.

### Methods

A cross-sectional study was conducted from January to June 2023 among 310 elderly patients (aged ≥60 years) selected through simple random sampling from both in-patient and out-patient departments. Polypharmacy was defined as the concurrent use of ≥5 medications, and excessive polypharmacy as ≥9 medications. Drug interaction screening was performed using UpToDate and Medscape interaction checker tools. Statistical analyses included descriptive statistics, t-tests, chi-square tests, and regression analysis, with significance set at $p < 0.05$.

**Data availability statement:** All relevant data are within the paper and its Supporting Information files.

**Funding:** The authors express their sincere thanks and gratitude to the AlMaarefa High Impact Research Support Program under Researchers Supporting Project number MHIRSP-2025001, AlMaarefa University, Riyadh, Saudi Arabia, for supporting the publication of this article.

**Competing interests:** The authors have declared that no competing interests exist.

## Results

The prevalence of polypharmacy was slightly higher in females (53.84%) than males (51.94%), with the highest rates observed in the 70–79 age group (48.88%). Most patients experienced moderate (50%) drug interactions. A significant gender difference was observed in the number of drugs prescribed at treatment ($p = 0.0032$), but not at discharge. Regression analysis identified gender ($p = 0.018$) and inpatient status ($p < 0.001$) as significant predictors of polypharmacy, while age was not ($p = 0.719$).

## Conclusion

This study reveals a high prevalence of polypharmacy among elderly patients attending Yenepoya Medical College and Hospital. Polypharmacy was found to be more prevalent among female patients, older age groups, and in inpatient wards. The distribution of drug interactions revealed a pervasive nature across various degrees, with moderate interactions being the most common. Future research should use larger samples, and diverse populations to better understand polypharmacy, drug interactions, and outcomes.

## Introduction

Polypharmacy, typically defined as the concurrent administration of five or more medications, has emerged as a significant concern in healthcare, particularly among the geriatric population [1]. With the increasing prevalence of chronic diseases and the corresponding rise in the number of prescribed medications, the complexities and challenges associated with polypharmacy have become more pronounced [2,3]. Globally, the prevalence of polypharmacy among older people ranges from 37% to 59%, varying by region and healthcare setting [4]. In India, studies have reported a prevalence rate of approximately 18% to 49% among the older people [5,6].

Polypharmacy poses considerable risks to older people, including adverse drug reactions (ADRs), drug interactions, medication non-adherence, and increased healthcare costs [7,8]. Moreover, these adverse effects can lead to a decline in physical and cognitive function, an increased risk of falls and hospitalization, and a reduced quality of life. Consequently, there is a critical need for comprehensive research to understand the consequences of polypharmacy and develop effective strategies to optimize medication management in this vulnerable population [9].

The World Health Organization (WHO) defines polypharmacy as the administration of an excessive number of drugs concurrently, often quantified as five or more medications [10]. While the simultaneous use of multiple medications may be necessary to manage various health conditions, it can also lead to inappropriate prescribing practices, where patients are prescribed more medications than clinically warranted [11]. This phenomenon, known as potentially inappropriate prescribing (PIP), encompasses overprescribing, mis-prescribing, and under-prescribing [12]. Inappropriate polypharmacy not only fails to address patients' clinical needs effectively, but also increases the risk of adverse outcomes, particularly in older adults [13].

A significant concern associated with polypharmacy is the increased risk of ADRs [14]. Adverse reactions can occur due to various mechanisms, including the pharmacological effects of the drugs, individual patient characteristics, and drug-drug interactions [15]. Although some ADRs result directly from the intended pharmacological action of the medication, others are unpredictable and may not be related to the drug's mode of action [16]. Furthermore, older people are particularly vulnerable to ADRs due to age-related physiological changes that affect the pharmacokinetics and pharmacodynamics of drugs, making them more susceptible to adverse effects [17].

Inappropriate polypharmacy and its associated adverse outcomes have prompted researchers to explore factors contributing to this phenomenon [18]. Studies have identified several determinants of polypharmacy, including the presence of multiple chronic conditions, depressive symptoms, and prescribing practices [19]. Physicians play a crucial role in the development of inappropriate polypharmacy, as highlighted by research emphasizing the importance of clinical decision-making, patient education, and communication in medication management [8].

Additionally, variations in polypharmacy rates across different healthcare settings underscore the need for tailored interventions to address this issue effectively [20]. Public primary care practices often exhibit higher rates of polypharmacy compared to private practices, suggesting the influence of healthcare system factors on prescribing patterns [21]. Addressing these disparities requires a multifaceted approach that considers the unique needs and challenges of diverse patient populations.

Moreover, existing research on polypharmacy often focuses on drug-drug interactions between two medications and lacks comprehensive assessments of pharmacogenomic issues, plasma drug concentrations, and clinical outcomes. Therefore, this study aimed to determine the prevalence and patterns of polypharmacy and pDDIs among elderly patients attending Yenepoya Medical College and Hospital, Mangalore, India.

## Methodology

This cross-sectional study was carried out among older patients at Yenepoya Medical College, a tertiary care teaching hospital in Karnataka, India. The hospital caters to a diverse patient population from urban, suburban, and rural regions, providing a suitable setting for examining the impacts of polypharmacy in the geriatric population. All study subjects had provided informed written consent before the study began.

### Ethical approval and consent

Ethical clearance was obtained from the Institutional Review Board (IRB) of Yenepoya Medical College and Hospital prior to the commencement of the study (Ethical approval number: YEC-1/2022/041). Informed consent was obtained from all participants or their legally authorized representatives before enrolling in the study. The study was conducted in accordance with the principles outlined in the Declaration of Helsinki, and patient confidentiality was strictly maintained throughout the research process.

### Sample size and sampling technique

The required sample size was determined using the Raosoft online sample size calculator, assuming a 5% margin of error, 95% confidence interval, and a 50% response distribution to minimize selection bias. The estimated sample size was 310 participants.

A simple random sampling technique was used to select eligible participants from both in-patient (IP) and out-patient (OP) departments, ensuring equal representation across clinical units.

### Study population and definitions

This study classified elderly patients as persons who were 60 years of age or older. Polypharmacy was defined as the simultaneous administration of five or more medications during a particular clinical interaction, specifically the initial visit

between January and June 2023. Excessive polypharmacy is characterized by the concurrent use of nine or more drugs. Topical drugs, including nasal drops, ear drops, emergency injections, and inhalers, were not included in the tally of routine prescriptions. The examination did not encompass herbal treatments, non-prescription medications, and vitamins. The medical diagnoses were categorized based on the 10th version of the International Classification of Diseases (ICD-10).

## Inclusion and exclusion criteria

Inclusion criteria encompassed older people aged 60 years and above who sought medical care at Yenepoya Medical College and Hospital. Only the data from the initial visit were acquired from the electronic records if the patient had several visits throughout the chosen time frame. The exclusion criteria were critically ill patients requiring intensive care, patients with severe cognitive impairment or mental retardation, and individuals unwilling to participate in the study.

## Data collection procedure

Data collection was carried out systematically over six months (from January 9, 2023, to June 22, 2023) to ensure completeness and consistency. The process was conducted by trained research personnel, including nurses and medical assistants, under the supervision of the principal investigator.

**Step 1: Patient screening and recruitment.** All patients aged 60 years and above attending the IP and OP departments of Yenepoya Medical College and Hospital were screened according to the inclusion and exclusion criteria. Eligible participants were approached during their hospital visit or admission. The study objectives, procedures, potential risks, and benefits were clearly explained to them in their preferred language. Written informed consent was obtained from all participants or their legally authorized representatives prior to enrolment.

**Step 2: Data source and extraction.** The data were obtained entirely from patients' electronic medical records (EMR) and hospital case files. No direct interviews were conducted to minimize recall bias and maintain data accuracy. The medical records provided detailed information on demographic characteristics, diagnosis, comorbidities, and all prescribed medications.

A standardized data collection form was designed to ensure uniformity and completeness. The form included sections for:

• **Demographics:** age, gender, and admission status (inpatient/outpatient).

• **Medical History:** major comorbidities such as hypertension, diabetes mellitus, hyperlipidemia, chronic kidney disease, and cerebrovascular accidents.

• **Medication Details:** complete drug list including generic name, dosage, frequency, route of administration, and duration of therapy.

Only medications prescribed at the initial clinical encounter during the study period were recorded. Topical preparations (e.g., eye/ear drops, nasal sprays), emergency drugs, vitamins, herbal, and over-the-counter medications were excluded to ensure uniform assessment.

**Step 3: Identification of potential drug–drug interactions.** For each participant, the complete medication list was entered and analyzed using two widely recognized clinical decision support tools: Medscape Drug Interaction Checker and UpToDate (Lexicomp). Both platforms provide comprehensive, evidence-based databases for identifying and categorizing potential drug–drug interactions (pDDIs).

Each identified interaction was classified according to severity levels (minor, moderate, or major) based on the software's clinical interpretation. The pDDIs were assessed at two time points:

1. At the initiation of treatment, and

2. At discharge (for inpatients), to evaluate any change in interaction risk.

**Step 4: Data coding and quality assurance.** After extraction, all data were cross-verified by two independent reviewers to ensure accuracy and resolve discrepancies. Data was coded numerically for statistical analysis. Age was categorized into five groups (60–65, 66–70, 71–75, 76–80, and ≥81 years). Polypharmacy was defined as the use of five or more medications, and excessive polypharmacy as nine or more concurrent drugs.

All data were anonymized to maintain patient confidentiality. Quality checks were performed periodically to ensure data completeness and reliability before statistical analysis.

## Statistical analysis

Descriptive statistics were performed. Categorical variables were computed for frequencies and percentages, while means and standard deviations were computed for continuous variables. T-tests and chi-square tests were employed to assess associations between the variables and drug responses. Regression analysis and correlation analyses were conducted to evaluate the relationships between independent and dependent variables. All statistical analyses were conducted using appropriate software packages such as SPSS or R, with statistical significance set at $p < 0.05$.

## Results

The study analysed data from a total of 310 elderly patients who were admitted to the Inpatient and attended the Outpatient departments of Yenepoya Medical College and Hospital. The demographic characteristics of the study population are summarized in Table 1. The mean age of the study participants was 70.25 years, with a standard deviation of 5.56 years. Out of the 310 participants, there was a higher representation of male participants compared to females. The majority of the participants fell within the age group of 66–70 years of the total sample. This was followed by the age groups of 71–75 years, 60–65 years, 76–80 years, and 81 years and above, respectively. A significant proportion of the study participants were admitted to the Inpatient ward, while the remaining sought medical care from the Outpatient department.

Table 2 shows prevalence of polypharmacy varied by gender and age. Among females, it was slightly higher than the prevalence in males. Similarly, the 70–79 age group revealed higher prevalence rates compared to other age groups.

Fig 1 presents the distribution of potential drug–drug interactions (pDDIs) according to their severity classification. Moderate interactions represented the largest proportion, occurring in 155 patients (50%). Severe interactions were identified in 88 patients (28.4%), whereas mild interactions were observed in 67 patients (21.6%). These findings suggest that while moderate interactions were most common, a notable proportion of patients were exposed to severe interactions, emphasizing the importance of careful medication review in patients receiving multiple therapies.

Table 1. Demographic characteristics of study participants.

| Variables | Frequency (n) | Percentage (%) |
|---|---|---|
| **Age group** | | |
| 60–65 years | 69 | 22.3 |
| 66–70 years | 111 | 35.8 |
| 71–75 years | 75 | 24.2 |
| 76–80 years | 43 | 13.9 |
| 81 and above | 12 | 3.9 |
| **Gender** | | |
| Male | 188 | 60.6 |
| Female | 122 | 39.4 |
| **In-Patient** | 267 | 86.1 |
| **Out-Patients** | 43 | 13.9 |

**Table 2. Prevalence of polypharmacy by gender and age.**

| Variables | Count | Prevalence (%) | 95% CI |
|---|---|---|---|
| **Gender** | | | |
| Female | 122 | 53.846154 | 44.27 - 63.43 |
| Male | 188 | 51.948052 | 44.06 - 59.84 |
| **Age** | | | |
| 50-59 | 1 | 100.000000 | 100.00 - 100.00 |
| 60-69 | 147 | 45.522388 | 37.09 - 53.95 |
| 70-79 | 140 | 48.888889 | 40.46 - 57.32 |
| 80-89 | 21 | 52.380952 | 31.02 - 73.74 |
| 90-99 | 1 | 0.000000 | 0.00 - 0.00 |

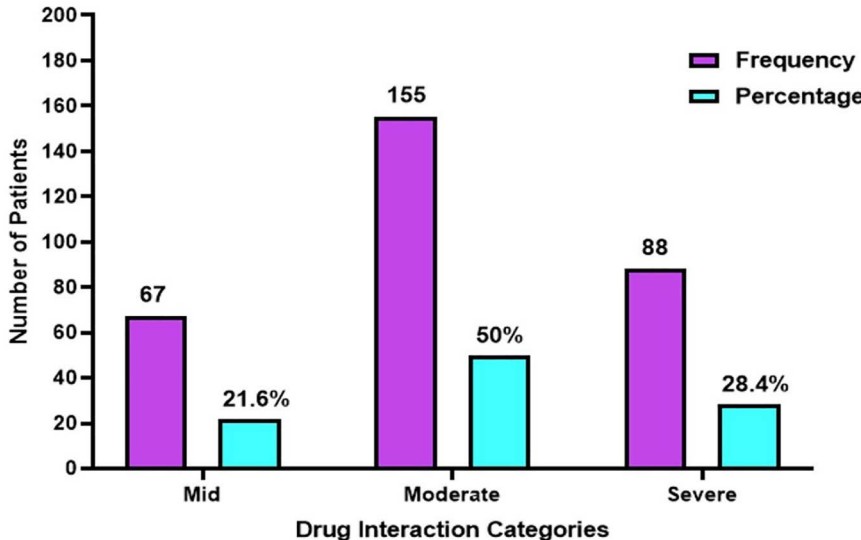

**Fig 1. The distribution of patient's drug interactions categorized by severity.**

Fig 2 illustrates the distribution of drug interactions based on polypharmacy status, defined as the concurrent use of five or more medications. Patients receiving ≥5 medications demonstrated substantially higher frequencies of drug interactions across most interaction categories compared with those receiving fewer than five medications. The greatest number of interactions among patients with polypharmacy was recorded at six interactions (52 patients), followed by three interactions (40 patients) and four interactions (34 patients). Conversely, patients receiving <5 medications exhibited comparatively lower interaction frequencies, primarily concentrated within zero and one interaction categories. These results highlight a significant relationship between polypharmacy and increased occurrence of drug interactions.

Fig 3 depicts the pattern of drug interactions among patients with advanced polypharmacy, defined as the use of more than nine medications. Patients receiving >9 medications showed markedly elevated interaction frequencies compared with those prescribed fewer medications. The highest interaction frequency was observed at six interactions (46 patients), followed by three interactions (29 patients) and four interactions (26 patients). In contrast, patients receiving <9 medications demonstrated consistently lower interaction counts. This pattern indicates that the likelihood and magnitude of drug interactions increase with escalating medication burden.

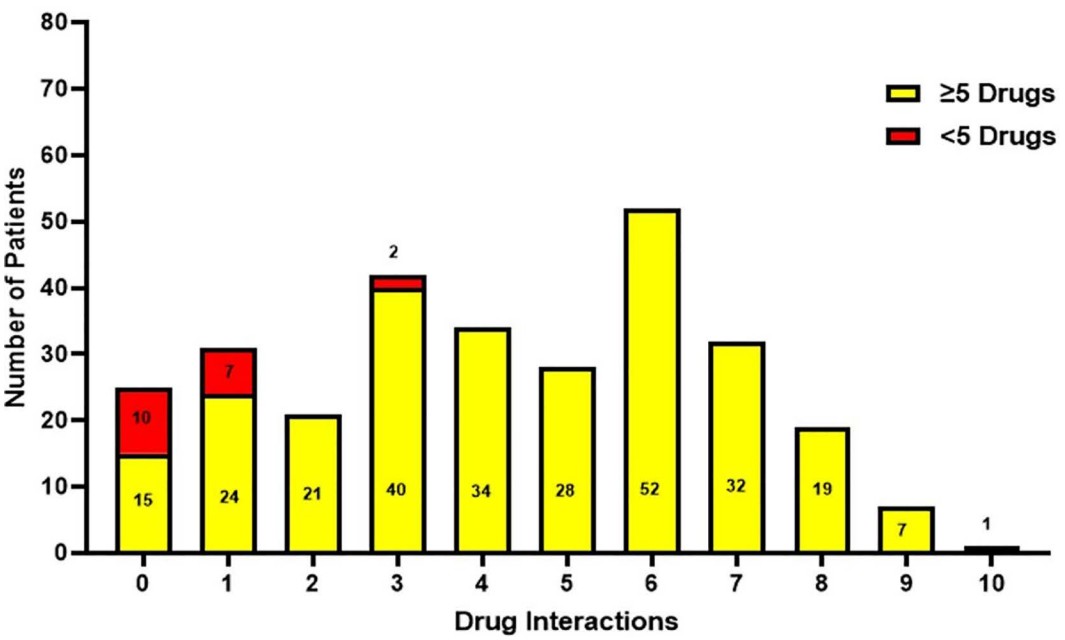

**Fig 2. Drug interaction with polypharmacy greater than or equal 5 drugs.**

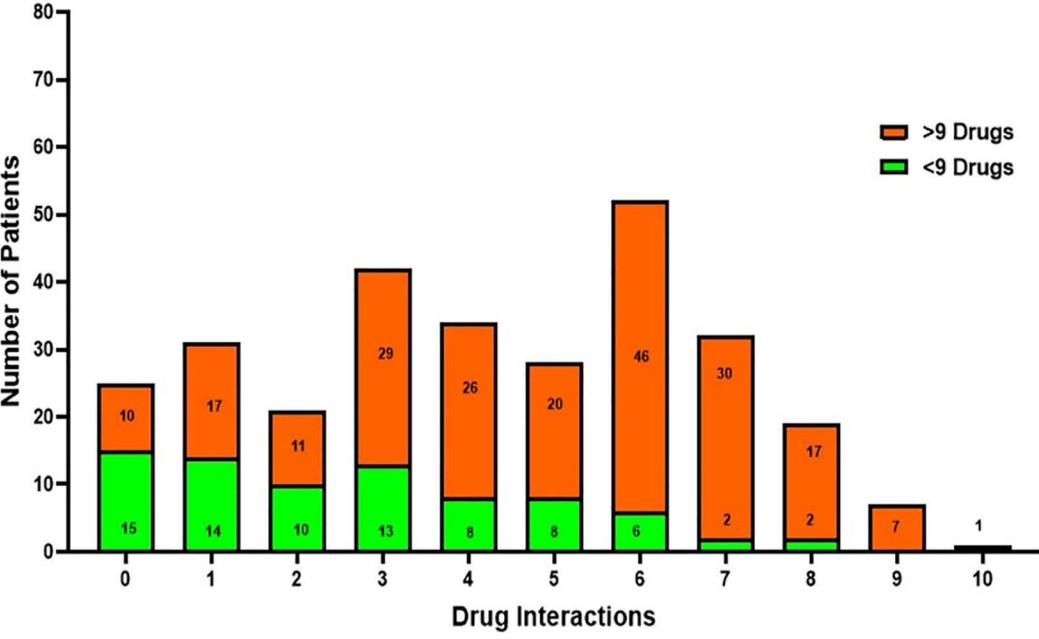

**Fig 3. Drug interactions with polypharmacy greater than 9 drugs.**

Fig 4 demonstrates the distribution of commonly prescribed medications used for managing comorbid conditions within the study population. Pantoprazole was the most frequently prescribed medication (n = 222), followed by aspirin (n = 98). Metformin and furosemide were prescribed equally (n = 85 each), while atorvastatin was prescribed to 81 patients. Other frequently used medications included dexamethasone (n = 66), gabapentin combined with nortriptyline (n = 59), tramadol and liquid paraffin (n = 54 each), and pramipexole and fluconazole (n = 38 each). The diversity of prescribed medications reflects the complexity of managing patients with multiple comorbidities and highlights the potential for increased drug interaction risk.

Fig 5 displays the frequency distribution of the most frequently identified potential drug–drug interaction pairs. The most commonly observed interaction involved metoprolol and timolol (11.2%), followed by nifedipine with amlodipine (9.35%) and levofloxacin with ondansetron (8.06%). Additional notable interaction pairs included azithromycin with heparin (5.41%), sodium bicarbonate with levofloxacin (5.16%), and quetiapine with pramipexole (2.9%). Several other drug combinations were observed with lower frequencies. These findings identify specific medication pairs that may require enhanced clinical monitoring to prevent adverse outcomes.

Table 3 shows the age and number of drugs prescribed at treatment and discharge. The correlation coefficient between age and the number of drugs prescribed at treatment indicates a strong positive correlation between these two variables. In this case, a coefficient close to 1 suggests that as age increases, the number of drugs prescribed at treatment tends to increase as well. Based on the chi-squared test results, no statistical significance reported between age and number of drugs prescribed at treatment and discharge.

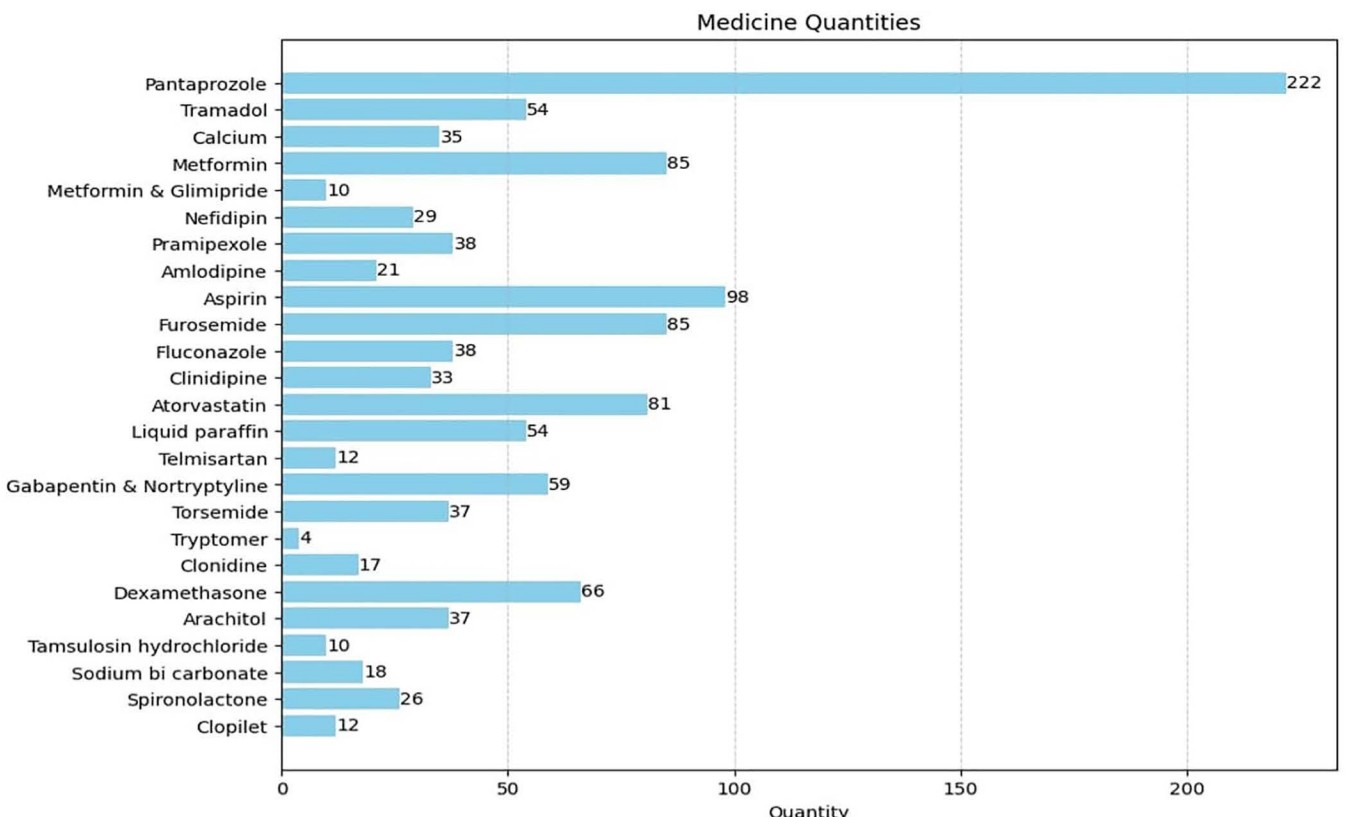

**Fig 4. Specific drugs prescribed for the management of various co-morbidities.**

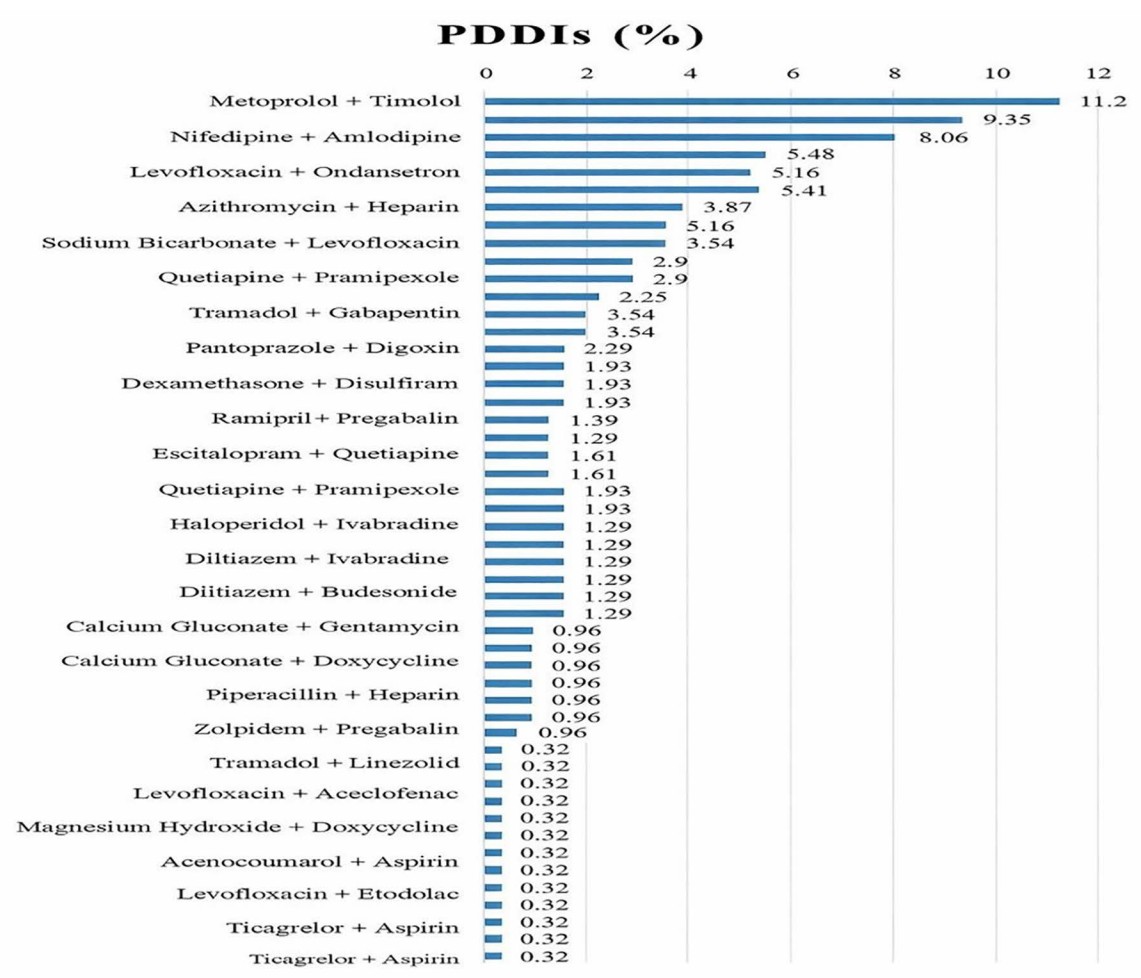

**Fig 5. Frequency of most common pDDIs administered among patients.**

**Table 3. Age and the number of drugs prescribed at treatment and discharge.**

| Variables | Correlation coefficient | Chi-squared | Degrees of freedom (df) | p-value |
|---|---|---|---|---|
| At Treatment | 0.8978203 | 536.2 | 486 | 0.05718 |
| At Discharge | 0.7532042 | 447.88 | 486 | 0.8915 |

Significant value <0.05; Correlation test; Chi-square test.

Table 4 shows t-test analysis. The result revealed a significant difference in the number of drugs prescribed at treatment between genders. However, no significant difference was observed at discharge.

Table 5 shows age vs mild, moderate, and severe drug interactions. The results of the one-way ANOVA test for the mild, moderate, and severe variable suggest that there is no statistically significant difference in the mean age based on the presence or absence of drug interactions.

Table 6 shows gender vs mild, moderate, and severe drug interactions. The Two Sample t-tests revealed no statistically significant difference between the mild, moderate, severe drug interactions and gender.

**Table 4. Gender and the number of drugs prescribed at treatment and discharge.**

| Variables | t-test | p-value |
|---|---|---|
| At Treatment | −2.99 | 0.0032 |
| At Discharge | 0.5331 | −0.62 |

Significant value <0.05; t-test.

**Table 5. Age vs mild, moderate, and severe drug interactions.**

| Variables | Degrees of freedom (df) | Sum of Squares (Sum Sq) | Mean Square (Mean Sq) | F value | p-value (Pr (>F)) |
|---|---|---|---|---|---|
| **Mild** | 1 | 0.99 | 0.994 | 0.046 | 0.839 |
| **Residuals** | 5 | 108.43 | 21.687 | | |
| **Moderate** | 8 | 262 | 32.72 | 1.064 | 0.39 |
| **Residuals** | 195 | 5994 | 21.687 | | |
| **Severe** | 6 | 121 | 20.18 | 0.417 | 0.828 |
| **Residuals** | 95 | 4070 | 42.85 | | |

Significant value <0.05; ANOVA test.

**Table 6. Gender vs mild, moderate, and severe drug interactions.**

| Variables | t | 95% CI | Degrees of freedom (df) | p-value |
|---|---|---|---|---|
| Mild | 1.3147 | −0.0501 - 0.2515 | 235.37 | 0.1899 |
| Moderate | −0.17263 | −0.6423 - 0.5388 | 232.3 | 0.8631 |
| Severe | −1.1092 | −0.4821 - 0.1352 | 165.02 | 0.269 |

Significant value <0.05; Two sample t test.

The regression analysis (Table 7) reveals several key findings regarding the factors influencing the number of drugs prescribed. The intercept indicates that, on average, 8.89 drugs are prescribed when all other variables are held constant. Age does not significantly impact the number of drugs prescribed. Gender is a significant predictor, with females being prescribed more drugs compared to males. The status of being an IP or OP is also significant, with OPs receiving fewer medications than IPs. Additionally, the presence of drug interactions significantly increases the number of drugs prescribed. The model explains approximately 35% of the variability in the number of drugs prescribed, as indicated by the R-squared value of 0.350. Table 8 shows the most commonly occurring drug-drug interactions.

**Table 7. Association between polypharmacy and predictors at Yenepoya Medical College and Hospital.**

| Predictors | Coefficient (β) | Standard Error (SE) | t-value | p-value | 95% Confidence Interval (CI) |
|---|---|---|---|---|---|
| **Intercept** | 8.8909 | 2.499 | 3.558 | 0.000 | 3.972, 13.810] |
| **Age** | 0.0125 | 0.035 | 0.360 | 0.719 | [-0.056, 0.081] |
| **Gender** | 0.9803 | 0.413 | 2.371 | 0.018 | [0.167, 1.794] |
| **IP/OP** | −4.6676 | 0.670 | −6.968 | 0.000 | [-5.986, -3.349] |
| **Drug interactions** | 0.5353 | 0.085 | 6.310 | 0.000 | [0.368, 0.702] |

**Table 8. Potential drug-drug interactions.**

| Drug combinations | PDDIs (%) | Clinical types of PDDIs | Mechanism of PDDIs | Potential risk |
|---|---|---|---|---|
| Metoprolol + Timolol | 35 (11.29%) | Severe | PK | Both increase anti-hypertensive blocking channel |
| Nifedipine + Tolvaptan | 29 (9.35%) | Severe | PK | Increases the level of tolvaptan by affecting hepatic enzyme metabolism |
| Nifedipine + Amlodipine | 25 (8.06%) | Severe | PD | Increases the level of amlodipine by affecting hepatic enzyme metabolism |
| Sodium Bicarbonate + Levofloxacin | 17 (5.48%) | Severe | PD | Decreases the level of levofloxacin by inhibition of GI absorption |
| Levofloxacin + Ondansetron | 16 (5.16%) | Severe | PK and PD | Increases Qtc interval |
| Fludrocortisone + Tolvaptan | 14 (5.41%) | Severe | | Fludrocortisone decreases the level of tolvaptan by p-glycoprotein efflux transporter |
| Azithromycin + Heparin | 12 (3.87%) | Severe | PD, Synergism | Increases the effect of heparin by decreasing metabolism |
| Dexamethasone + Ivabradine | 11 (3.54%) | Severe | PD, Antagonism | Decreases the effect of ivabradine by affecting hepatic enzyme metabolism |
| Sodium Bicarbonate + Levofloxacin | 11 (3.54%) | Severe | PD, Antagonism | Sodium bicarbonate decreases the level of levofloxacin by inhibition of GI absorption |
| Ceftriaxone + Calcium Acetate | 9 (2.90%) | Severe | PD, Antagonism | Calcium salts enhance toxic effect of ceftriaxone |
| Quetiapine + Pramipexole | 9 (2.90%) | Severe | PD, Synergism | Pharmacodynamic synergism |
| Ceftriaxone + Enoxaparin | 7 (2.25) | Severe | PD, Antagonism | Increases the effect of enoxaparin by anticoagulation |
| Tramadol + Gabapentin | 6 (1.93%) | Moderate | PD | enhances CNS depressant effect of tramadol |
| Tramadol + Desloratadine | 6 (1.93%) | Moderate | PK | Enhances CNS depressant effect of tramadol |
| Pantoprazole + Digoxin | 6 (1.93%) | Moderate | PK | Increases the level of digoxin by increasing gastric pH |
| Clopidogrel And Aspirin + Pantoprazole | 5 (1.61%) | Moderate | PD | Decreases serum conc. Of active metabolite of clopidogrel |
| Dexamethasone + Disulfiram | 5 (1.61%) | Moderate | PK | Disulfiram may enhance the toxic effect of dexamethasone |
| Clonidine + Metoprolol | 5 (1.61%) | Moderate | PD, Synergism | Pharmacodynamic synergism |
| Ramipril + Pregabalin | 4 (1.29%) | Moderate | PD, Synergism | Pharmacodynamic synergism |
| Spironolactone + Potassium Chloride | 4 (1.29%) | Moderate | PD | Increases serum potassium |
| Escitalopram + Quetiapine | 4 (1.29%) | Moderate | PK | Increases toxicity of quetiapine by QTc interval |
| Haloperidol + Pramipexole | 4 (1.29%) | Moderate | PD, Antagonism | Pharmacodynamic antagonism |
| Quetiapine + Pramipexole | 5 (1.61%) | Moderate | PD, Antagonism | Pharmacodynamic antagonism |
| Quetiapine + Levodopa | 5 (1.61%) | Moderate | PD, Antagonism | Pharmacodynamic antagonism |
| Haloperidol + Ivabradine | 4 (1.29%) | Moderate | PK | Increases the level of ivabradine by affecting hepatic enzyme metabolism |
| Ranolazine + Metformin | 4 (1.29%) | Moderate | PD, Antagonism | Increases the effect of metformin by decreasing the elimination |
| Diltiazem + Ivabradine | 4 (1.29%) | Moderate | PD, Synergism | Increases the level of ivabradine by affecting hepatic enzyme metabolism |
| Hydrocortisone + Ranolazine | 4 (1.29%) | Moderate | PK | Decreases the level of ranolazine by affecting hepatic enzyme metabolism |
| Diltiazem + Budesonide | 4 (1.29%) | Moderate | PD | Increases the level of budesonide by affecting hepatic enzyme metabolism |
| Budesonide + Spironolactone | 4 (1.29%) | Moderate | PK and PD | Decreases the level of spironolactone affecting hepatic enzyme metabolism |

*(Continued)*

Table 8. (Continued)

| Drug combinations | PDDIs (%) | Clinical types of PDDIs | Mechanism of PDDIs | Potential risk |
|---|---|---|---|---|
| Calcium Gluconate + Gentamycin | 4 (1.29%) | Moderate | PD, Synergism | Pharmacodynamic synergism |
| Torsemide + Gentamycin | 3 (0.96%) | Moderate | PD, Synergism | Pharmacodynamic synergism |
| Calcium Gluconate + Doxycycline | 3 (0.96%) | Moderate | PD, Antagonism | either decreases the level of other by inhibition of GI absorption |
| Ceftriaxone + Heparin | 3 (0.96%) | Moderate | PK and PD | Ceftriaxone increases the level of heparin by anticoagulation |
| Piperacillin + Heparin | 3 (0.96%) | Moderate | PD, Antagonism | Piperacillin increases the level of heparin by anticoagulation |
| Doxycycline + Ivabradine | 2 (0.64%) | Mild | PK and PD | Doxycycline increases the level of ivabradine by affecting hepatic enzyme metabolism |
| Zolpidem + Pregabalin | 2 (0.64%) | Mild | PD, Antagonism | Pregabalin enhances CNS depressant effect of zolpidem |
| Spironolactone + Potassium Chloride | 1 (0.32%) | Mild | PD, Antagonism | Both increases serum potassium |
| Tramadol + Linezolid | 1 (0.32%) | Mild | PD, Antagonism | Linezolid enhances the serotonergic effect of tramadol, which results in serotonin syndrome |
| Tramadol + Morphine | 1 (0.32%) | Mild | | Morphine increases CNS depressant effect of tramadol |
| Levofloxacin + Aceclofenac | 1 (0.32%) | Mild | PD, Antagonism | Aceclofenac increases the neuroexcitatory effect of levofloxacin |
| Mirtazapine + Azithromycin | 1 (0.32%) | Mild | PD, Antagonism | Both increases QTc interval |
| Magnesium Hydroxide + Doxycycline | 1 (0.32%) | Mild | PK and PD | Magnesium hydroxide decreases the level of doxycycline by inhibition of GI absorption |
| Doxycycline + Amoxicillin | 1 (0.32%) | Mild | PK and PD | Pharmacodynamic antagonism |
| Acenocoumarol + Aspirin | 1 (0.32%) | Mild | PD, Antagonism | Aspirin enhances the anticoagulant effect of acenocoumarol |
| Clopidogrel + Pantoprazole | 1 (0.32%) | Mild | PK and PD | Pantoprazole decreases serum conc. Of active metabolite of clopidogrel |
| Levofloxacin + Etodolac | 1 (0.32%) | Mild | PD, Antagonism | Etodolac enhances neuroexcitatory effect of levofloxacin |
| Amitriptyline + Ondansetron | 1 (0.32%) | Mild | PD, Antagonism | Both increase sedation/ either increases toxicity of other by serotonin level |
| Ticagrelor + Aspirin | 1 (0.32%) | Mild | PK and PD | Aspirin increases antiplatelet effect of ticagrelor |

## Discussion

In this study, a total of 340 elderly patients were screened, of which 310 met the inclusion criteria and were enrolled. The medical records of individuals aged 60 years and above—often presenting with multiple comorbidities—were meticulously examined to determine the prevalence of polypharmacy and potential drug–drug interactions (pDDIs). Polypharmacy, defined as the concurrent use of multiple medications, remains a significant concern in geriatric care due to its implications for treatment complexity, adverse drug events, and healthcare costs [22]. The present findings revealed a high prevalence of polypharmacy among elderly patients, underscoring the multifaceted challenges associated with managing medications in older adults with multiple chronic conditions.

The study population had a mean age of 70.25 years, reflecting the growing demographic of older adults who are particularly vulnerable to medication-related problems. This observation aligns with global trends indicating a rapid increase in the elderly population, accompanied by a rise in multimorbidity and medication dependence [23,24]. Focusing on this

population is crucial, as aging is associated with physiological alterations that affect pharmacokinetics and pharmacody-namics, increasing susceptibility to adverse drug reactions and interactions [25].

A slight predominance of male participants was observed in this study, which may reflect gender-based differences in healthcare-seeking behavior, disease prevalence, and access to healthcare facilities. Prior research has documented similar gender disparities, highlighting that women often engage more proactively with healthcare services but may also exhibit higher medication use due to chronic disease prevalence [26,27]. The majority of participants were recruited from in-patient settings, where the likelihood of polypharmacy and pDDIs is higher due to intensive therapeutic interventions and the management of multiple comorbidities. Hospitalized elderly patients typically receive more medications than those in out-patient care, reflecting the complexity and acuity of their conditions [28].

In the current study, females exhibited a slightly higher prevalence of polypharmacy compared to males. This finding is consistent with previous studies by Hosseini et al. (2018) [27] and Pereira et al. (2017) [29], which also reported higher polypharmacy rates among elderly females. This gender difference may stem from the higher burden of chronic condi-tions, such as osteoporosis, arthritis, and hypertension, among women, necessitating multiple medications. The preva-lence of polypharmacy among individuals aged 60–69 years was 45.5%, increasing to 48.9% in the 70–79 age group. These rates align with the global prevalence range of 30–60% reported by the United Nations (2020) [23] and corrobo-rate findings from Young et al. (2021) [30] and Midão et al. (2018) [31], both of which noted that polypharmacy tends to increase with advancing age due to the accumulation of chronic conditions and therapeutic regimens.

The present study also revealed that most drug interactions were of moderate severity (50%), followed by severe and mild categories. The predominance of moderate interactions suggests that while most pDDIs may not immediately threaten life, they still require careful monitoring and dose adjustments to prevent adverse outcomes. Only 21.6% of interactions were classified as mild, reflecting the clinical vigilance required in prescribing for elderly patients. These findings agree with studies by Noor et al. (2019) [32], Obeid et al. (2022) [33], and Admassie et al. (2013) [34], all of which observed moderate interac-tions as the most prevalent. In contrast, Rabba et al. (2020) [35] and Eneh et al. (2020) [36] reported a higher proportion of severe interactions, likely due to differences in study design, patient population, and classification criteria.

The pattern of drug interactions observed in our study mirrors prior findings emphasizing the strong correlation between polypharmacy and pDDIs among elderly populations [37,38]. Mohamed et al. (2023) [38] and Hermann et al. (2021) [39] demonstrated that an increasing number of prescribed drugs correlates with a higher likelihood of clinically significant interactions. Mohamed et al. (2023) [38], in particular, found that polypharmacy among older adults with advanced cancer was strongly associated with unfavorable treatment outcomes due to drug interactions. Similarly, Hermann et al. (2021) [39] documented high rates of polypharmacy and pDDIs among community-dwelling elderly patients, highlighting the need for medication review even outside hospital settings. These parallels reinforce our finding that higher medication counts substantially elevate the risk of pDDIs.

The correlation analysis between age and the number of prescribed drugs at treatment revealed a positive associa-tion, suggesting that medication burden increases with advancing age. However, the relationship did not achieve statis-tical significance, implying that comorbidities and treatment complexity, rather than age alone, are the primary drivers of polypharmacy. This finding aligns with the observations of Scondotto et al. (2018) [40] and other studies [33,39,41], which reported similar trends but emphasized that disease burden and healthcare utilization patterns play a more direct role than chronological age in determining medication load.

Interestingly, no significant association was found between age and the number of drugs prescribed at discharge. This may be due to standardized discharge protocols that emphasize deprescribing and medication optimization before dis-charge, a finding consistent with prior studies [39,40]. This underscores the importance of individualized medication recon-ciliation and comprehensive discharge planning to ensure continuity of care and minimize unnecessary drug exposure.

Gender differences in prescribing were also observed. Females received a significantly higher number of medications at treatment compared to males, consistent with the findings of Zucker and Prendergast (2020) [42] and Alwhaibi and

Balkhi (2023) [43]. However, this difference disappeared at discharge, likely reflecting standardized prescribing protocols that promote rationalization of therapy at discharge. Similar findings were reported by Glans et al. (2020) [44] and Parekh et al. (2020) [45], suggesting that while inpatient prescribing may vary by gender, discharge regimens are typically harmonized to ensure optimal medication use.

The literature consistently identifies factors such as age, comorbidities, duration of hospital stay, and polypharmacy as major risk determinants for pDDIs [45–47]. Elderly patients, in particular, are vulnerable due to physiological changes affecting drug metabolism and excretion, coupled with altered pharmacokinetics and pharmacodynamics. These changes heighten susceptibility to adverse drug events and therapeutic failures [48]. Recognizing these factors is essential for clinicians to mitigate risks and enhance patient safety.

Our study contributes to existing literature by reaffirming that while age and gender influence prescribing patterns, the primary determinant of pDDI risk remains the number of medications prescribed. Rodrigues et al. (2020) [49] similarly found that polypharmacy and physiological aging increase interaction risk, while Bories M, et al. (2022) [50] reported that when controlling for comorbidities and drug count, age alone is not a significant predictor of drug interactions.

The absence of significant gender differences in the severity of drug interactions in our study also aligns with findings by Seeman (2021) [51], suggesting that while pharmacokinetic and pharmacodynamic differences exist between sexes, these may not always translate into clinically significant differences in DDI severity.

Regression analysis identified gender and inpatient status as significant predictors of polypharmacy, whereas age was not. Females and inpatients were more likely to receive multiple medications, corroborating results from Manteuffel et al. (2014) [52] and Tian et al. (2021) [53]. The latter study highlighted that hospitalized patients, due to disease complexity and intensive therapeutic needs, often receive broader drug regimens. Furthermore, our results demonstrated that the presence of pDDIs itself predicted an increase in medication count, reflecting clinical attempts to manage adverse effects or drug inefficacies through additional prescriptions, an observation previously reported by Bucșa et al. (2013) [54].

Overall, the regression model explained 35% of the variance in drug count, suggesting that other unmeasured factors such as physician prescribing habits, patient adherence, and healthcare policy also contribute to polypharmacy patterns. Similar multifactorial influences were reported by Hsu et al. (2021) [55], emphasizing the interplay of clinical, behavioural, and systemic factors in shaping prescribing practices [56–66].

## Limitations

This cross-sectional study may be influenced by selection bias, confounding variables, and potential inaccuracies in medical records. The study's single-centre setting limits the generalizability of findings. Additionally, variations in the accuracy and comprehensiveness of drug interaction screening tools may affect the results.

## Recommendations

Future studies should include larger, multicentric samples and more diverse populations to better understand the relationship between polypharmacy, drug interactions, and clinical outcomes. Further, interventional studies focusing on optimizing medication management and minimizing polypharmacy-related risks are recommended to enhance geriatric patient care.

## Conclusion

This study reveals a high prevalence of polypharmacy among elderly patients attending Yenepoya Medical College and Hospital. Polypharmacy was found to be more prevalent among female patients, older age groups, and in inpatient wards. The distribution of drug interactions revealed a pervasive nature across various degrees, with moderate interactions being the most common. Our findings underscore the critical role of healthcare professionals in conducting comprehensive medication reviews and identifying potential pDDIs to ensure safe and effective pharmacotherapy.

Key Points

- Polypharmacy increases the incidence of Adverse Drug Reactions (ADRs), hospitalizations, and mortality in older adults.

- Managing multiple comorbidities often requires complex medication regimens, leading to a higher likelihood of potential drug-drug interactions (pDDIs).

- Altered pharmacokinetics and pharmacodynamics in elderly patients complicate drug metabolism and excretion, making them more prone to medication-related adverse effects.

- Polypharmacy can contribute to medication non-adherence, cognitive decline, and diminished functional independence.

- Clinicians are recommended to conduct meticulous evaluations of pharmacological protocols, judiciously curtail superfluous medications, mitigate pDDIs, and tailor therapeutic strategies to harmonize with patients' evolving clinical states and individualized health objectives.

## Supporting information

**S1 File. Anonymized data set polypharmacy.**
(XLSX)

**S2 File. Data statistics.**
(DOCX)

**S3 File. Raw data for graphs.**
(CSV)

## Acknowledgments

The authors express their thanks and gratitude to AlMaarefa University, Riyadh, Saudi Arabia for providing support to publish this article.

## Author contributions

**Conceptualization:** Rokeya Sultana.

**Data curation:** Rokeya Sultana, Ranajit Das.

**Formal analysis:** Umaima Farheen Khaiser, Ranajit Das.

**Funding acquisition:** Saeed G. Alzahrani, Mohammed Khairt Newigy, Mohammed Sharique Ahmed Quadri, Mujeeb Ahmed Shaikh, Farha M Shaikh, Mohammad Fareed.

**Investigation:** Umaima Farheen Khaiser, Rokeya Sultana.

**Methodology:** Umaima Farheen Khaiser, Rokeya Sultana, Madhumitha Kannan, Mohammad Fareed.

**Project administration:** Rokeya Sultana.

**Resources:** Rokeya Sultana, Saeed G. Alzahrani.

**Software:** Ranajit Das.

**Supervision:** Rokeya Sultana.

**Validation:** Umaima Farheen Khaiser, Saeed G. Alzahrani, Madhumitha Kannan, Mohammed Khairt Newigy, Mohammed Sharique Ahmed Quadri, Mujeeb Ahmed Shaikh, Farha M Shaikh, Mohammad Fareed.

**Visualization:** Rokeya Sultana.

**Writing – original draft:** Umaima Farheen Khaiser, Rokeya Sultana, Mohammad Fareed.

**Writing – review & editing:** Umaima Farheen Khaiser, Rokeya Sultana, Saeed G. Alzahrani, Madhumitha Kannan, Mohammed Khairt Newigy, Mohammed Sharique Ahmed Quadri, Mujeeb Ahmed Shaikh, Farha M Shaikh, Mohammad Fareed.

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
