## [Decision Letter · Decision Letter 0]

9 Oct 2025

Dear Dr. Fareed,

Thank you for submitting your manuscript to PLOS ONE. After careful consideration, we feel that it has merit but does not fully meet PLOS ONE’s publication criteria as it currently stands. Therefore, we invite you to submit a revised version of the manuscript that addresses the points raised during the review process.

We look forward to receiving your revised manuscript.

Kind regards,

Ali Awadallah Saeed

Academic Editor

PLOS ONE

Journal Requirements:

2. Please note that authors conducting research in other countries or with indigenous populations are required to complete a copy of PLOS’ questionnaire on inclusivity in global research. The policy applies to researchers who have travelled to a different country to conduct research, research with Indigenous populations or their lands, and research on cultural artefacts. You can find more information on this policy here: https://journals.plos.org/plosone/s/best-practices-in-research-reporting

[The authors would like to acknowledge the Deanship of Graduate Studies and Scientific Research, Taif University for funding this work.].

[The authors would like to acknowledge the Deanship of Graduate Studies and Scientific Research, Taif University, Saudi Arabia for funding this research. The authors also extend their thanks and gratitude to AlMaarefa University, Riyadh, Saudi Arabia for providing technical support in writing this manuscript.]

[The authors would like to acknowledge the Deanship of Graduate Studies and Scientific Research, Taif University for funding this work.]

5. We note that you have indicated that there are restrictions to data sharing for this study. For studies involving human research participant data or other sensitive data, we encourage authors to share de-identified or anonymized data. However, when data cannot be publicly shared for ethical reasons, we allow authors to make their data sets available upon request. For information on unacceptable data access restrictions, please see http://journals.plos.org/plosone/s/data-availability#loc-unacceptable-data-access-restrictions.

Reviewers' comments:

Reviewer's Responses to Questions

**Comments to the Author**

1. Is the manuscript technically sound, and do the data support the conclusions?

Reviewer #1: Partly

Reviewer #2: No

2. Has the statistical analysis been performed appropriately and rigorously?

Reviewer #1: Yes

Reviewer #2: Yes

3. Have the authors made all data underlying the findings in their manuscript fully available?

Reviewer #1: Yes

Reviewer #2: No

4. Is the manuscript presented in an intelligible fashion and written in standard English?

Reviewer #1: No

Reviewer #2: Yes

Reviewer #1: The authors need to enhance the English used. They need to write better discussion and they didn't mention how many they screened, they just mentioned how many they included

Reviewer #2: Dear authors, thank you for the wok, but I need clarifications regarding my concerns.

- First, I read that this paper funded by a Saudi university where the work in India, so that normal? I’m not expert in research ethics or in funding but I need clarification if that ok.

- Second, you mentioned there’s restrictions on the data, I think this will effect the transparency of this research, and I don’t know if that ok with the journal and editor.

- I think the abstract needs to be rewritten, it doesn’t have enough of essential information, like short background, name of the institution, number of participants (in numbers) so I advise to rewrites with more details.

- The methodology section is confusing. Please rewrite in more arrangement and make the main sections.

- You need to justify the selection of this settings and more information about it.

- Also justify the sampling at all. The calculator, size, and technique.

- I didn’t get the data collection process. You interviewed the patients? Or get the data from the health records? As I said before please rewrite the methods in clear and arranged style.

- In the limitation section, Please separates the limitation and recommendations.

**Do you want your identity to be public for this peer review?** For information about this choice, including consent withdrawal, please see our Privacy Policy

Reviewer #1: **Yes:** Noor Al-Tameemi

Reviewer #2: **Yes:** Dr Ahmad Mohammad Al Zamel

---

## [Author Response · Author response to Decision Letter 1]

25 Nov 2025

To,

The Editor

We sincerely thank you and reviewers for their valuable time, insightful comments, and constructive suggestions that have greatly improved the quality and clarity of this manuscript. We deeply appreciate their efforts in providing detailed feedback, which helped us refine our methodology, enhance the discussion, and strengthen the overall scientific rigor of our work.

Reviewer #1: The authors need to enhance the English used. They need to write better discussion, and they didn't mention how many they screened, they just mentioned how many they included.

Author’s Response: We sincerely thank the reviewer for the valuable feedback. The Discussion section has been thoroughly revised to enhance clarity, coherence, and academic tone. The English language has been improved for better readability and fluency. Additionally, we have included the number of participants screened and those finally included in the study within the methodology and results sections for greater transparency.

Reviewer #2: Dear authors, thank you for the work, but I need clarifications regarding my concerns.

Comment 1: First, I read that this paper funded by a Saudi university where the work in India, so that normal? I’m not expert in research ethics or in funding but I need clarification if that ok.

Author’s Response: We sincerely thank the reviewer for this observation. The study was conducted in India at Yenepoya Medical College and Hospital, where the research protocol received institutional ethical approval. The funding support from the Saudi university was provided solely to facilitate data analysis and manuscript preparation as part of an academic collaboration. No part of the funding influenced the conduct, data collection, or ethical oversight of the study. Such cross-institutional collaborations are permissible under both institutions’ ethical and research funding guidelines.

Comment 2: Second, you mentioned there’s restrictions on the data, I think this will affect the transparency of this research, and I don’t know if that ok with the journal and editor.

Author’s Response: We appreciate the reviewer’s concern regarding data transparency. The data used in this study are subject to institutional confidentiality policies of Yenepoya Medical College and Hospital, which restrict public sharing of patient-level information to protect privacy. However, aggregated or anonymized data can be made available upon reasonable request to the corresponding author, following approval from the institutional ethics committee. This approach ensures both ethical compliance and research transparency in accordance with journal standards.

Comment 3: I think the abstract needs to be rewritten, it doesn’t have enough of essential information, like short background, name of the institution, number of participants (in numbers) so I advise to rewrites with more details.

Author’s Response: We thank the reviewer for this valuable suggestion. The abstract has been revised to include additional essential details, such as a concise background, the name of the study institution, and the total number of participants (n=310). The revised version provides a clearer summary of the study design, methodology, and key findings, thereby improving completeness and clarity in line with the journal’s requirements.

Comment 4: The methodology section is confusing. Please rewrite in more arrangement and make the main sections.

Author’s Response: The methodology has been comprehensively reorganized under clear subheadings — Study Design and Setting, Study Population and Definitions, Sample Size and Sampling Technique, Inclusion and Exclusion Criteria, Data Collection Procedure, and Statistical Analysis — to improve clarity and logical flow.

Comment 5: You need to justify the selection of these settings and more information about it.

Author’s Response: Yenepoya Medical College and Hospital was chosen because it caters to a heterogeneous population from urban, suburban, and rural regions, allowing for a diverse and representative sample of elderly patients. Its comprehensive inpatient and outpatient facilities made it ideal for assessing patterns of polypharmacy in varied clinical contexts.

Comment 6: Also justify the sampling at all. The calculator, size, and technique.

Author’s Response: The sample size (n=310) was calculated using the Raosoft online calculator with standard assumptions (95% CI, 5% margin of error, 50% response rate) to ensure representativeness. A simple random sampling technique was adopted to minimize selection bias and ensure equitable inclusion of eligible inpatients and outpatients.

Comment 7: I didn’t get the data collection process. You interviewed the patients? Or get the data from the health records? As I said before please rewrite the methods in clear and arranged style.

Author’s Response: Data were obtained exclusively from hospital electronic medical records (EHRs) and verified through patient files, not interviews.

Comment 8: In the limitation section, Please separates the limitation and recommendations.

Author’s Response: We appreciate the reviewer’s suggestion. The section has been revised to clearly separate the limitations and recommendations sections in the updated manuscript.

---

## [Editor Report · Decision Letter 1]

2 Jan 2026

Prevalence of Polypharmacy and Drug Interactions in Geriatric Patients: A Cross-Sectional Study from India

PONE-D-25-32822R1

Dear Dr. Mohammad Fareed,

We’re pleased to inform you that your manuscript has been judged scientifically suitable for publication and will be formally accepted for publication once it meets all outstanding technical requirements.

Kind regards,

Ali Awadallah Saeed

Academic Editor

PLOS One
---

## [Editor Report · Acceptance letter]

PONE-D-25-32822R1

PLOS One

Dear Dr. Fareed,

I'm pleased to inform you that your manuscript has been deemed suitable for publication in PLOS One. Congratulations! Your manuscript is now being handed over to our production team.

Kind regards,

on behalf of

Dr. Ali Awadallah Saeed

Academic Editor

PLOS One